# Evaluation of Immune Infiltrates in Ovarian Endometriosis and Endometriosis-Associated Ovarian Cancer: Relationship with Histological and Clinical Features

**DOI:** 10.3390/ijms241512083

**Published:** 2023-07-28

**Authors:** Emanuela Spagnolo, Alejandra Martinez, Andrea Mascarós-Martínez, Josep Marí-Alexandre, María Carbonell, Eva González-Cantó, Eva Manuela Pena-Burgos, Bárbara Andrea Mc Cormack, Sarai Tomás-Pérez, Juan Gilabert-Estellés, Ana López-Carrasco, Paula Hidalgo, Martina Aida Ángeles, Andrés Redondo, Alejandro Gallego, Alicia Hernández

**Affiliations:** 1Department of Gynecology, La Paz University Hospital, 28046 Madrid, Spain; emanuelaspagnolo01@gmail.com (E.S.); mariacarbonell676@gmail.com (M.C.); analopezcarrasco.lopez@gmail.com (A.L.-C.); aliciahernandezg@gmail.com (A.H.); 2Research Institute “IdiPaz”, La Paz University Hospital, 28046 Madrid, Spain; 3Department of Surgical Oncology, Institut Claudius Regaud-Institut Universitaire du Cancer du Toulouse (IUCT) Oncopole, 31059 Toulouse, France; martinez.alejandra@iuct-oncopole.fr (A.M.); martinangeles22@hotmail.com (M.A.Á.); 4Centre de Recherches en Cancérologie de Toulouse, UMR 1037 INSERM, 31100 Toulouse, France; 5Department of Pathology, General University Hospital of Valencia, 46014 Valencia, Spain; andrea-a.m@hotmail.com; 6Research Laboratory in Biomarkers in Reproduction, Obstetrics and Gynecology, Research Foundation, General University Hospital of Valencia, 46014 Valencia, Spain; evagonzalezcanto@gmail.com (E.G.-C.); barbymccormack@gmail.com (B.A.M.C.); sarai.altea@gmail.com (S.T.-P.); juangilaeste@yahoo.es (J.G.-E.); 7Department of Pathology, La Paz University Hospital, 28046 Madrid, Spain; evita_gm_k11@hotmail.com; 8Department of Obstetrics and Gynecology, General University Hospital of Valencia, 46014 Valencia, Spain; 9Department of Paediatrics, Obstetrics and Gynaecology, University of Valencia, 46010 Valencia, Spain; 10Department of Radiology, La Paz University Hospital, 28046 Madrid, Spain; paulaa.hidalgo@salud.madrid.org; 11Department of Medical Oncology, La Paz University Hospital, 28046 Madrid, Spain; redondos@uam.es (A.R.); alex_gallego@hotmail.es (A.G.); 12Department of Obstetrics and Gynaecology, Universidad Autónoma Madrid, 28049 Madrid, Spain

**Keywords:** endometriosis, endometriosis-associated ovarian cancer, tumour-infiltrating lymphocytes, TIM3, FOXP3, PD-1, CD39, CD163 macrophages

## Abstract

Background: the association between ovarian endometriosis (OE) and endometriosis-associated ovarian cancer (EAOC) is extensively documented, and misfunction of the immune system might be involved. The primary objective of this study was to identify and compare the spatial distribution of tumour-infiltrating lymphocytes (TILs) and tumour-associated macrophages (TAMs) in OE and EAOC. Secondary objectives included the analysis of the relationship between immunosuppressive populations and T-cell exhaustion markers in both groups. Methods: TILs (CD3, CD4, and CD8) and macrophages (CD163) were assessed by immunochemistry. Exhaustion markers (PD-1, TIM3, CD39, and FOXP3) and their relationship with tumour-associated macrophages (CD163) were assessed by immunofluorescence on paraffin-embedded samples from *n* = 43 OE and *n* = 54 EAOC patients. Results: we observed a predominantly intraepithelial CD3+ distribution in OE but both an intraepithelial and stromal pattern in EAOC (*p* < 0.001). TILs were more abundant in OE (*p* < 0.001), but higher TILs significantly correlated with a longer overall survival and disease-free survival in EAOC (*p* < 0.05). CD39 and FOXP3 significantly correlated with each other and CD163 (*p* < 0.05) at the epithelial level in moderate/intense CD4 EAOC, whereas in moderate/intense CD8+, PD-1+ and TIM3+ significantly correlated (*p* = 0.009). Finally, T-cell exhaustion markers FOXP3-CD39 were decreased and PD-1-TIM3 were significantly increased in EAOC (*p* < 0.05). Conclusions: the dysregulation of TILs, TAMs, and T-cell exhaustion might play a role in the malignization of OE to EAOC.

## 1. Introduction

Endometriosis is a common oestrogen-dependent inflammatory disease that affects millions of women and teen girls worldwide, up to 50% of women with chronic pelvic pain, and 30–50% of women with infertility [1]. The endometriotic microenvironment is composed of an intricate mixture of distinct cell types (i.e., endometrial epithelial cells, stromal fibroblasts, and immune cells), metabolic waste products, and steroid hormones, among others. Remarkably, several components of this endometriotic microenvironment might favour the growth and development of ovarian cancer (OC) [2]. The association between ovarian endometriosis (OE) and OC is largely supported by extensive epidemiological studies which have coined the entity endometriosis-associated ovarian cancer (EAOC) [3]. EAOC is represented by epithelial ovarian tumours, mainly clear cell ovarian carcinomas (CCOC) and endometrioid ovarian carcinomas (EOC). In contrast to the most frequent and aggressive epithelial tumour of the ovary (high-grade serous ovarian cancer, HGSOC), EAOC presents several distinctive clinical characteristics, highlighting its usual diagnosis at early stages and the presence of mutations in *KRAS*, *ERBB2*, *PTEN*, *PIK3CA*, and *ARID1A* and rare mutations in *TP53* [4]. A large epidemiologic study from the Dutch National Database points to the association of endometriosis as a protective factor in OC patients. Specifically, the authors demonstrated that patients with OC and concomitant endometriosis have increased overall survival (OS) compared to those OC patients without concomitant endometriosis [5].

At the cellular level, the role of the immune system has attracted recent increasing research interest both in OC and in the malignant transformation of endometriosis [3]. Regarding OC, Santoiemma and colleagues [6] showed that infiltration by CD3+CD4+ and CD3+CD8+ T-cell tumour-infiltrating lymphocytes (TILs) played a crucial role in disease progression and were associated with a positive prognosis. Khalique and collaborators [7] performed a transcriptomic and immunofluorescence characterization of CCOC tumours, showing spatial differences between immunosuppressive immune and effector populations that could identify patients with a high risk of recurrence and those with a potential response to immune checkpoint therapy. Further research allowed the observation that endometriosis specimens may present cancer-like immune cell infiltrates. Edwards et al. [8] showed that patients with OE display either a benign inflammatory transcriptomic profile or activation of the complement pathway and humoral immunity. Notably, complement upregulation might be one of the key steps in the early carcinogenesis of patients with OE by favouring immunosuppression and neoangiogenesis. In contrast, Nero et al. [9] reported a specific T-cell immune pattern in EAOC. Specifically, compared to OE, EAOC presented lower TILs and higher programmed death-1/programmed death ligand-1 (PD-1/PD-L1) expression profiles. Interestingly, one-third of OE also had immune cancer-like infiltrate. These results suggest that early immune changes might define a group of OE patients with a high risk of developing EAOC, characterized by low intraepithelial CD3+CD4+ and/or CD3+CD8+ T-cell infiltration and T-cell exhaustion (the latter being in turn characterized by the expression of markers as CD39 and immune checkpoints as PD-1 and T-cell immunoglobulin and mucin-domain containing-3 (TIM3)). Nevertheless, the effect of an endometriosis-specific microenvironment and inflammation profile on immune cell compartment distribution and function remains unknown.

The primary objective of this study is to identify and compare the spatial distribution of CD3+/CD4+ and CD3+/CD8+ TILs and tumour-associated macrophages (TAMs) in OE and EAOC. Secondary objectives include the analysis of the relationship between immunosuppressive populations and T-cell exhaustion markers in both groups.

## 2. Materials and Methods

### 2.1. Study Cohort

We conducted a retrospective multicentre case-control study of patients with OE (study period 2016–2019) and EAOC (study period 1999–2019) who were diagnosed and treated at the La Paz University Hospital (Madrid, Spain) and the General University Hospital of Valencia (Valencia, Spain). A pathologist with more than 10 years of experience in Gynaecological Pathology confirmed the diagnosis of OE and/or EAOC at each institution, respectively. Demographic, clinicopathological, surgical, and follow-up data were collected. Research electronic data capture (REDCap, https://redcap.general-valencia.san.gva.es/redcap/, accessed on 20 March 2023) was used for optimal data collection and sharing between institutions [10].

Inclusion criteria comprised the following: patients of legal age, patients who signed the informed consent, patients undergoing surgery for symptomatic or suspected OE, and patients undergoing surgery for early or advanced EAOC with concomitant endometriosis. Exclusion criteria were patients with autoimmune diseases or a history of hepatitis B or C or human immunodeficiency virus. Treatment with gonadotropin releasing hormone agonists or oral contraceptives were not considered exclusion criteria. The revised classification of the American Fertility Society (rAFS) was used for endometriosis patients [11]. In the case of advanced EAOC, exploratory laparoscopy was performed to assess resectability (primary cytoreductive surgery) and to obtain a histological diagnosis. Patients with deep infiltration of the small bowel mesentery, diffuse carcinomatosis, and a tumour involving large parts of the small bowel, stomach, and infiltration of the duodenum or pancreas were considered unresectable and selected for neoadjuvant chemotherapy. All patients underwent cytoreductive surgery after neoadjuvant chemotherapy. The tumour residual after the cytoreductive procedure was measured using the Completeness of Cytoreduction score [12].

Disease-free survival (DFS) was defined as the time (months) from the surgery date to the first recurrence or last follow-up. OS was defined as the time (months) from diagnosis to all causes of death (or the date of the last follow-up for alive patients). Recurrent disease was identified by clinical symptoms (pelvic pain and weight loss), elevation of tumour blood markers (CA-125 levels), and suggestive imaging findings during follow-up.

Ethical statement: all patients were included in the study after signing informed consent. The research was carried out following the ethical principles of the Declaration of Helsinki (1964) and its subsequent modifications [13]. The study was approved by the Clinical Research Ethics Committee of the La Paz University Hospital of Madrid (PI-3350) and the General University Hospital of Valencia (FIHGUV 2020/200).

### 2.2. Samples Preparation

Endometriotic and tumoral formalin-fixed lesions were paraffin-embedded. Based on Hematoxylin and Eosin stains performed for routine pathological diagnosis, the block that best represented the heterogeneity of the lesion and its microenvironment for each patient was defined. The selected block was cut into serial 4 μm sections, mounted onto glass slides, deparaffinized with xylene, and rehydrated with ethanol. The distribution of TILs and exhausted lymphocytes and macrophages was assessed in the whole tissue of each slide.

### 2.3. Immunohistochemistry

Immunohistochemistry (IHC) stainings with the antibodies of interest were carried out to determine the TILs population and distribution. Specific markers of T lymphocytes (CD3: IR503; CD4: IR649; and CD8: IR 623 (Dako-, Agilent, Santa Clara, CA, USA)) were employed, and analyses were performed using Autostainer Link 48 automated equipment (Dako-Agilent, Santa Clara, CA, USA). For the label determination, appropriate peroxidase-conjugated secondary antibodies (Goat anti-Mouse IgG, #AP124P, (Merck, Darmstadt, Germany); Goat anti-Rabbit #AP132P, (Merck, Darmstadt, Germany) were used. Peroxidase activity was revealed with diaminobenzidine (ThermoScientific, Waltham, MA, USA). The immune marker expression was evaluated for the epithelial and stromal compartments at 10 independently selected areas at 40× magnification. Areas of 0.0625 mm^2^ were digitally photographed and counted manually. All sections were evaluated by two independent and experienced pathologists (J.M.-A. and E.M.P.-B.) who were blinded to the clinicopathologic parameters and clinical outcomes of the patients. For disagreements, the opinion of a third pathologist (A.M.-M.) was requested to define the case. Staining intensity was evaluated in each specimen. As a positive control of the technique, slices of a human amygdala without disease were used. The results were manually scored on a semi-quantitative scale of 0 to 3: 0, negative; 1, weak; 2, intermediate; and 3, intense signal.

### 2.4. Immunofluorescence

The distribution of exhausted lymphocytes and macrophages was assessed by immunofluorescence (IF). Specifically, those patients with moderate/intense signals of intra epithelial and/or stromal CD3+CD4+ and CD3+CD8+ TILs were selected. For patients with CD3+CD8+ moderate/intense signal, immune staining with specific exhausted lymphocytes markers PD-1 (1:100; ab214421, (Abcam, Cambridge, UK)) and TIM3 (1:50; PA5-18470 (Life Technology S.A. Invitrogen, Waltham, MA, USA)) were performed. In patients with CD3+CD4+ moderate/intense signal, Forkhead Box P3 (FOXP3) (1:200; PA1-9044 (Life Technology S.A. Invitrogen, Waltham, MA, USA)) and CD39 (1:100; PA5-97709 (Life Technology S.A. Invitrogen, Waltham, MA, USA)) antibodies were used. Additionally, distribution of CD163+ TAMs (1:100; ab87099, (Abcam, Cambridge, UK)) was examined in both groups. For the fluorimetric detection, the following secondary antibodies were used: Alexa Fluor^®^ 488 AffiniPure Fab Fragment Goat Anti-Rabbit IgG, Fc fragment specific (1:500; 111-547-008, (Jackson ImmunoResearch, Baltimore Pike, PA, USA)) for PD-and CD39; Alexa Fluor^®^ 594 AffiniPure Mouse Anti-Goat IgG (H+L) (1:500; 205-585-108 (Jackson ImmunoResearch, Baltimore Pike, PA, USA)) for TIM3 and FoxP3; and Alexa Fluor^®^ 647 AffiniPure Fab Fragment Goat Anti-Rabbit IgG, Fc fragment-specific (1:500; 111-607-008, (Jackson ImmunoResearch, Baltimore Pike, PA, USA)). Nuclear staining was carried out with 4′,6-diamidino-2-phenylindole (DAPI) (Merck, Darmstadt, Germany).

From the histological sections of the patients under study, IF assays were carried out for two sets of markers: FOXP3-CD39-CD163 (Set 1) and PD-1-TIM3-CD163 (Set 2). Firstly, the channel corresponding to each fluorophore was identified from the images obtained in a fluorescence microscope (DMI 4000 D (Leica, Wetzlar, Germany). Once the channels were determined, the mean intensity per channel/fluorophore was assessed in each patient’s slide using the imageJ program (U. S. National Institutes of Health, Bethesda, MD, USA, https://imagej.nih.gov/ij/, 1997–2018, accessed on 20 March 2023). The intensities of each marker were normalized to the corresponding DAPI intensity per photo. To determine if the patients had high or low intensity values for each marker, the median of each patient for a certain marker was compared with the group median of the same marker, thus determining if its value was above (HIGH) or under (LOW). Likewise, the average of the medians of the Set 1 and Set 2 markers was calculated, and the average of each patient was compared for the set to determine if it was HIGH or LOW. All subsequent statistical comparisons were made from these values.

### 2.5. Statistical Analysis

All the variables were checked for normality of the distribution using the Kolmogorov–Smirnov test. Qualitative data descriptions were made using absolute frequencies and percentages, and quantitative data descriptions were made using the mean ± the standard deviation of the mean or the median and interquartile range, depending on the adjustment to normality. Comparisons were made using the chi-square test or Fisher’s exact test for categorical variables and the Student’s *t*-test or Mann–Whitney test for independent data as parametric and non-parametric tests, respectively. Statistical significance levels for the correlations between quantitative variables were calculated using the Spearman’s correlation test. The survival analysis was carried out using the Kaplan–Meier analysis and the log-rank test. All statistical tests were considered bilateral and significant with *p*-values < 0.05. Data were analysed using R software (version 3.6.2) (The R foundation, Wien, Austria).

## 3. Results

During the study period, 97 patients met the inclusion criteria. Depending on their histopathology results, the patients were divided into two groups: OE patients (*n* = 43) and EAOC patients with concomitant endometriosis (*n* = 54). The EAOC group was further divided into two subgroups: endometrioid ovarian cancer (EOC; *n* = 30) and clear cell ovarian cancer (CCOC: *n* = 24). The presence of endometriosis in all cases of EAOC was histologically confirmed. The mean age of the patients with OE was 38.4 (± 6.1) years old, and the mean body mass index (BMI) (kg/m^2^) was 24.7 ± 3.5 kg/m^2^. Following endometriosis rAFS classification, 7 (16.28%) patients belonged to stage I-II; 26 (60.47%) belonged to stage III; and 10 (23.26%) belonged to stage IV. The mean age of the EAOC patients was 53.5 years old ±11.7, with a mean BMI of 25.8 ± 4.1 kg/m^2^. Patients with EAOC were significantly older than patients with OE (*p* < 0.001). Concerning BMI, no statistically significant differences were observed between the OE and EAOC groups (*p* = 0.17). The EOC and CCOC patients’ characteristics and follow-ups are summarized in Table 1. The median overall follow-up (months) for the EAOC patients was 87.50 (Q1–Q3: 35.5–156.5).

### 3.1. Immunohistochemical Analysis of TILs Distribution in OE and EAOC

The distribution of TILs in the OE, EAOC, and EAOC subgroups are shown in Table 2. Regarding the distribution of TILs, we observed that OE mainly presented an intraepithelial CD3+ distribution (80%), whereas EAOC showed predominantly both an intraepithelial and stromal pattern (62.5%) (*p* ≤ 0.001). Furthermore, the TILs signal was higher in OE than in EAOC (moderate/intense vs. low/mild, respectively; *p* ≤ 0.001). We did not find statistically significant differences in the TILs’ locations and signal characteristics within the different EAOCs subtypes.

### 3.2. Oncological Outcomes Depending on TILs Infiltration Signal in EAOC Patients

EAOC patients with TILs signal in the intratumoral and/or stromal compartments were included in the survival analyses. Survival outcomes and TIL infiltration patterns are summarized in Table 3. We observed a statistically significant association between a higher TILs infiltrate and a longer OS, OS in patients with adjuvant therapy, and DFS in patients with adjuvant therapy (*p* ≤ 0.03). Regarding Kaplan–Meier survival curves, TILs were associated with improved DFS in EAOC (*p* < 0.03), mainly in EOC (*p* = 0.018). In the CCOC subtype, TILs did not correlate with increased DFS (*p* = 0.261) (Figure 1). To confirm whether the observed associations were mainly due to the difference in TILs signal between groups, we compared clinical variables (age, FIGO stage, grade, surgical performance, and neoadjuvant and adjuvant treatment) between patients with low/mild and moderate/intense TILs signal in all EOAC patients and in the EOC and CCOC subgroups. No statistically significant differences were observed in any of these variables, discarding a confounding effect of these variables on our results.

### 3.3. Analyses of T-Cell Exhaustion Markers in EAOC

Immunofluorescence staining for T-cell exhaustion and TAMs markers was performed in those EAOC specimens in which CD4+ and/or CD8+ showed a moderate/intense signal in immunochemistry analyses (*n* = 13). Specifically, the FOXP3+/CD39+/CD163+ panel was measured in those patients with CD4+ moderate/intense signal and the PD-1+/TIM3+/CD163+ panel in those with CD8+ moderate/intense signal. At the intraepithelial level, we observed a statistically significant association in the expression of PD-1+ and TIM3+ (Pearson’s r = 0.96, *p* = 0.009), CD39+ and FOXP3+ (Spearman’s ρ = 0.83, *p* = 0.015), and CD39+ and CD163+ (Spearman’s ρ = 0.76, *p* = 0.037). At the stromal level, TIM3+ expression showed a tendency towards a positive correlation with the expression of CD163+ (Spearman’s ρ = 0.87, *p* = 0.056).

### 3.4. Comparison of T-Cell Exhaustion Markers in EAOC vs. OE

Regarding the expression of T-cell exhaustion and TAMs markers, FOXP3+/CD39+/CD163+ was significantly decreased in EAOC compared to OE in those patients with CD4+ moderate/intense signal. In addition, we observed a significant increase in PD-1+/TIM3+/CD163+ in EAOC compared to OE in those patients with CD8+ moderate/intense signal (Figure 2 and Figure 3).

## 4. Discussion

The association between OE and EAOC is extensively documented. Although the exact pathogenic mechanism of malignization remains elusive, a misfunction of the immune system might be involved. In this retrospective case-control study, we showed that OE presented with higher TILs infiltrate and different spatial TILs distribution compared to EAOC. Whereas OE displayed a predominant CD3+ intraepithelial distribution and higher CD3+ signal, EAOC showed a combined intraepithelial and stromal pattern, and mild/low CD3+ signal. OE also showed a significantly increased CD8+ infiltrate and a higher CD8+ signal compared to neoplastic lesions. T-cell subtype infiltrate as well spatial T-cell distribution, immune checkpoint expression profile, and co-localization with regulatory T-cells and myeloid suppressive populations in patients with moderate/intense CD4+ and/or CD8+ signals were suggestive of an inflamed phenotype and excluded profile in EAOC [14]. Regulatory and suppressive populations were enriched in OE lesions, putatively contributing to promoting the progression of EAOC.

In this study, we identified a significantly higher rate of moderate/intense TILs signal in OE. Similarly, results were reported in a retrospective study of Nero et al. [9] that included 55 EAOC patients and 55 OE patients. A significantly higher count of infiltrating T lymphocytes was observed in endometriosis cases compared to EAOC, mainly endometrioid. Furthermore, Scheerer et al. [15] showed that a specific subgroup (CD3+, CD4+, and CD8+ phenotype) of infiltrated immune cells was associated with ovarian and peritoneal endometriosis.

In our series, we identified differences in the spatial locations of CD3+ cells. In OE, CD3+ infiltrate was mainly located in the intraepithelial compartment, while CD3+ cells were located both in the intraepithelial and stromal compartment in 62.5% and 2% of EAOC and OE, respectively (*p* < 0.001). These results suggest stromal interactions leading to an excluded phenotype in EAOC. Other studies using the same methodological approach have also shown exclusion mechanisms in CCOC, particularly in patients with *ARID1A* mutation (*ARID1A*mut) [7,16] In particular, Devlin et al. [16] showed that patients with CCOC and *ARID1A* wild type (*ARID1A*wt) contained significantly higher amounts of CD3+, CD8+, and CD4+ cells in the stromal area compared to *ARID1A*mut tumours. Consequently, *ARID1A*mut CCOC showed a reduction in the infiltration and proliferation of T-cells alongside a reduction in stromal CD8+ and CD4+ T-cell activity. These authors hypothesized that *ARID1A*mut CCOC arise from endometriosis since *ARID1A*mut is an early event in the malignant transformation of OE [16]. Similarly, we observed stromal CD3+ and CD8+ signal reduction in CCOC compared to EOC, although without reaching statistical significance. In addition, in a cohort of 33 CCOC, Khalique et al. [7] found that CD8+ cells were more prevalent in the stroma of *ARID1A*mut patients. On the contrary, they failed to observe any differences in terms of CD4+ spatial distribution between *ARID1A*mut and *ARID1A*wt cases.

In our series, we did not observe any significant differences in terms of T-cell infiltrate and the intensity of signals and spatial distributions between EAOC subtypes. Nevertheless, EAOC showed a high percentage of CD3+ (98%), CD4+ (86.7%), and CD8+ (69.4%) signals (Table 2), confirming the “inflamed set up” of this malignancy [14]. In favour of this topic, in a study by Tuan et al. [17] that analyzed the gene expression profiles of 222 CCOC, the authors showed that the mesenchymal-like subtype was associated with a high enrichment of TILs, in particular CD4+. Our results agree with those from Howitt et al. [18], who documented no significant differences in intraepithelial TILs distribution in CCOC and HGSOC.

On the contrary, in a study by Milne et al. [19], intraepithelial lymphocytes were more prevalent in the endometrioid histotype in comparison to clear cell histotype. The survival benefits of TILs in EOC have been noted for a long time. In particular, the presence of TILs within the tumour microenvironment is considered to be an indication of the host immune response to tumour antigens [20], with favourable prognostic factors in epithelial ovarian cancer [21]. For instance, Murakami et al. [22] confirmed the positive effect of TILs expression in DFS y OS.

In the present study, patients with moderate/intense CD8+ signal showed co- expression of PD-1+ and TIM3+ at the intraepithelial level (*p* = 0.009) and significant intraepithelial co-expression of CD39+ FOXP3+ (*p* = 0.015), both correlating with enriched CD163+ cells. In contrast, TIM3+ and CD163+ showed a tendency towards a positive correlation in the stromal compartment (*p* = 0.056). A recent study by Khalique et al. [7] analyzed the spatial locations of immune subpopulations promoting an immunosuppressive environment and *ARID1A* mutational status in OCCC. The authors showed that PD-L1+ FOXP3+ CD4+ (T-regulatory) cells coexisted with CD68+ and PD- L1+ CD68+ tumour-associated macrophages (TAMs) in the stroma of *ARID1A*mut low-risk patients. The authors claimed that the ‘tumour-exclusion’ of these cells is important for maintaining an effective anti-tumour immune response and preventing tumour progression.

Additionally, Webb et al. [23] showed that CD103+ CD8 TILs expressed PD-1 and appeared quiescent in the tumour microenvironment. They speculated that, after standard treatment, CD8+ PD-1+ TILs regain functional anti-tumour activity. HGSOC tumours harboured a higher mean number of PD-1+ cells compared to endometrioid or clear cell tumours. In another study, Webb et al. [24] analyzed PD-L1 and other TILs markers in tissue microarrays containing the main histotypes of epithelial ovarian cancer (i.e., HGSOC, endometrioid, clear cell, and mucinous ovarian cancer). They found that PD-L1+ was associated with CD8+ TILs expressing PD-1+, CD103+, and FOXP3+. In particular, HGSOC patients presented a higher rate of PD-1+CD8+ (57.4%) in comparison to EOC and CCOC (22.4% and 16.2%, respectively). Surprisingly, in HGSOC, PD-L1+ was associated with a favourable prognosis. Indeed, PD-L1 can be expressed by tumour cells or macrophages, leading to reduced T-cell activity. Hamanishi and colleagues [25] showed a correlation between tumour PD-L1 expression and a lower intraepithelial CD8+ TIL count in HGSOC. In this way, PD-L1 promotes tumour immune escape and induces the functional impairment of tumour-specific T-cells. The authors explained this result with the concept of *adaptive resistance*, meaning that activated T-cells induced a negative feedback mechanism in the tumour microenvironment, resulting in an immunological stalemate.

The results of the current study confirmed the presence of a subset of exhausted T-cells in EAOC which regarding OC had only been described in the high-grade serous subtype, to the best of our knowledge. This might represent an opportunity to select those patients which could benefit the most from immunotherapy. Regarding EAOC, this could become an especially attractive therapeutic opportunity for CCOC, the subtype of OC that is more resistant to conventional chemotherapy.

In contrast, patients with CCOC had a two-fold higher response rate in the KEYNOTE 100 study which assessed the performance of pembrolizumab (a monoclonal IgG4 antibody drug against anti–PD-1) in patients with advanced recurrent ovarian cancer [26]. A single institution retrospective series showed clinical durable response to immune checkpoint blockers treatment in 25% of patients with CCCO. Notably, respondent patients presented with high PD-1 TILs infiltrate [27]. In our series, we observed a higher expression of PD-1 in EAOC in comparison to OE. Nero et al. [9] also revealed higher levels of PD-1/PD-L1 expression in EAOC in contrast to OE. The authors hypothesized that decreasing TILs and increasing PD-1/PD-L1 could be significant steps of the pathological transformation from endometriosis to EAOC.

Next, we showed the enrichment of exhausted T-cells, with TIM3+ and PD-1+ expression in the intraepithelial compartment of EAOC with high CD8+ infiltrate (*p* < 0.05). This result is in line with those from Sawada et al. [28], who analyzed the expression of PD-1+ and TIM3+ on CD8+ T-cells in 100 ovarian cancer patients. The authors observed that a high percentage of PD-1+ TIM3+ in ovarian cancer samples was associated with poor patient prognosis. In another study by Balança et al. [20], the authors showed that TIM3 was the last checkpoint to be acquired and was systematically associated with PD-1 expression. The co-expression of PD-1 and TIM3 in T- cells was a surrogate marker of T-cell tumour-antigen specificity [4].

In EAOC, we also found that intratumoral CD39+ and FOXP3+ were positively correlated in patients with high CD4+ T-cell infiltrate. Accordingly, Balança et al. [29] identified a population of CD4+ T-cells defined by high PD-1+, CD39+, and FOXP3- in the head and neck of cervical and OC patients. CD39+ and PD-1+ co-expression in CD4+ T-cells were surrogate markers of CD4+ T-cells specificity. Other studies have also shown different FOXP3+ populations, with specific subtypes specialized in type I T-cells responses. Indeed, CXC3+ Tregs have been found in ovarian cancers, where they are directly correlated with effector cells and constitute the main Treg population [30].

In endometriosis, many studies report enhanced Tregs resulting in local immune suppression and induced lesion development and growth [2,31,32]. It has been reported that the proportion of CD4+CD25highFOXP3+ Treg cells may be significantly increased in the peritoneal fluid of patients with endometriosis [33]. Several studies have shown a strong association between tumour infiltration by FOXP3+ CD4+ TILs and endometrioid subtype [19,34]

Regarding CD163+ TAMs, we observed a predominant distribution both at the intraepithelial and stromal level of EAOC in our series. Wei et al. [4] observed that TAMs are a key component of tumour stroma and of neoplasm progression, blocking TILs action. Single cell studies also reveal the highest degree of CD8+ T-cell exhausted populations and CD163+ macrophages in ovarian cancer with an infiltrated immune phenotype [35], speculating that suppressive macrophages could contribute to lesion growth and progression to EAOC. The function of suppressive macrophages contributes to the survival of ectopic endometrial tissue by inducing immune tolerance and stimulating angiogenesis [2,36].

In our series, we observed a statistically significant association between greater TILs infiltrate and better OS and DFS in patients with EAOC (*p* = 0.01). In particular, in the endometrioid subgroup, TILs infiltrate significantly correlated with DFS (*p* < 0.005). Similar results were shown by Gallego et al. [37], who demonstrated that high levels of intraepithelial CD8+ TILs were associated with longer survival in EOC. 

Furthermore, intratumoral CD3+ and CD8+ tumour-infiltrating lymphocytes showed an improved prognosis in the endometrioid subtype. In addition, in a recent article involving 1078 EOC and 545 CCOC, a significant association was observed for disease-specific survival in EOC in a univariable analysis [38]. However, the largest series analyzing the relationship between TILs and survival included HGSOC [39]. These studies evaluated different types of infiltrating T-cells. In particular, a higher count of CD8+ was associated with a better prognosis [40]. Treg TILs were associated with either a negative [41] or a positive impact [42]. Finally, Yildirim et al. [43] showed that CD3+ and CD8+ T lymphocyte infiltrations were related to advanced stage, high-grade ovarian cancer and poor prognosis. In our study, we showed TILs moderate/intense signal and longer OS in patients with complete surgery performance. In this regard, several studies showed that TILs were more prevalent in optimally debulked patients compared to those with macroscopic residual disease [6,19,28].

### Limitations and Strengths of the Study

The main limitations of this study are its retrospective design and sample size. Given the lower frequency of EAOC, with respect to HGSOC, the participation of a large number of centers is required to obtain an adequate number of samples. Nevertheless, we presented a comprehensive data collection of 43 patients with OE and 54 patients with EAOC and concomitant endometriosis that represents a strength of our study. The strengths are the originality in the type of cohort, with samples of endometriosis and EAOC, the homogenous treatment of these patients, and the clinical, survival, and immunochemistry data. Another strength of our study is its potential applicability regarding a disease of great prevalence and social, occupational, and personal repercussions, such as endometriosis.

In this article, the immunologic results of women with OE and EAOC were presented. The survival value of TILs infiltrated in the endometrioid ovarian cancer subtype represents an interesting and encouraging result to further investigate the possible use of immunotherapy in these subtypes of ovarian cancer. However, we believe that the best clinical approach should consider both the immunologic findings and the genetic characteristics of the patients. As one of the possible limitations of this work, we did not incorporate genetic assessments of the included patients, which will be considered in future publications.

## 5. Conclusions

Ovarian endometriosis and endometriosis-associated ovarian cancer share similar immunologic profiles. However, a significantly higher rate of moderate/intense TILs signals were shown in OE.

Exhaustion pathways showed PD1+TIM3+ have a prevalent intraepithelial distribution in EAOC.

High Treg levels were expressed in OE, showing the possible role of these regulatory cells in endometriosis progression to OC. CD163+ macrophages (TAMs) have a key role and may contribute, in collaboration with Tregs lymphocytes, to endometriosis growth and the development of EAOC.

Finally, regarding EAOC, TILs infiltrate significantly correlated with DFS.

## Figures and Tables

**Figure 1 ijms-24-12083-f001:**
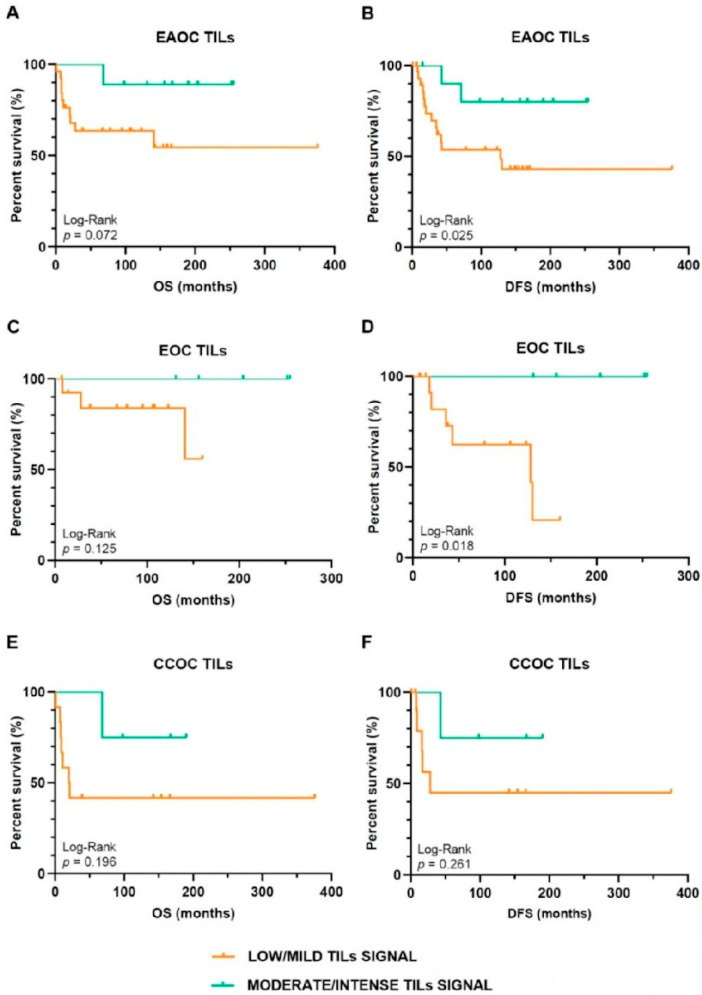
Kaplan–Meier survival curves. Comparison of: (**A**) DFS and (**B**) OS in EAOC patients; (**C**) OS and (**D**) DFS in EOC patients; and (**E**) OS and (**F**) DFS in CCOC, attending to TILs signal intensity (low/mild vs. moderate/intense). CCOC: clear cell ovarian cancer; DFS: disease-free survival; EAOC: endometriosis-associated ovarian cancer; EOC: endometrioid ovarian cancer; OS: overall survival; TILs: tumour-infiltrating lymphocytes.

**Figure 2 ijms-24-12083-f002:**
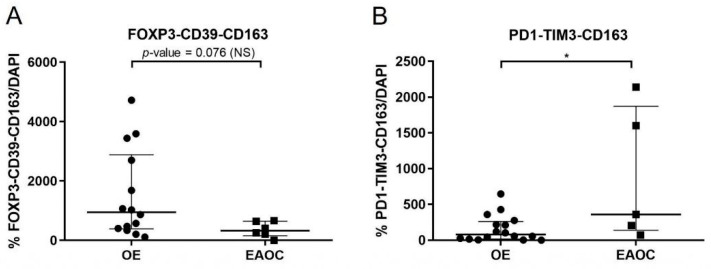
Comparison of immunofluorescence results in EAOC vs. OE for the two subsets of markers analyzed: (**A**) FOXP3-CD39-CD163 and (**B**) PD1-TIM3-CD163. In all cases, peritumoral and intraephitelial staining were considered together. * *p*-value < 0.05. EAOC: endometriosis-associated ovarian cancer; NS: no significance; OE: ovarian endometriosis.

**Figure 3 ijms-24-12083-f003:**
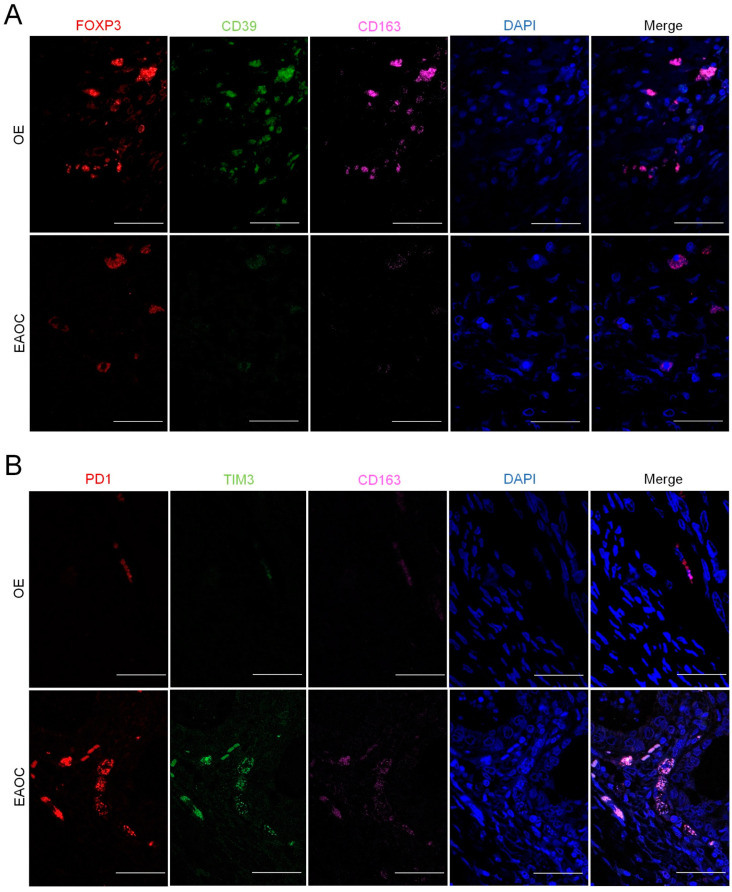
Representative examples of staining (**A**) FOXP3-CD39-CD163-DAPI-Merge and (**B**) PD-1-TIM3-CD163-DAPI-Merge in tissue samples of patients with OE and EAOC. EAOC: endometriosis-associated ovarian cancer; OE: ovarian endometriosis. Scale bar: 50 µm.

**Table 1 ijms-24-12083-t001:** Clinical and demographic characteristics of the following subgroups. EAOC: EOC vs. CCOC.

	EOC (*n* = 30)	CCOC (*n* = 24)	*p*-Value
Age (years) (mean ± SD)	52.0 ± 10.3	55.4 ± 13.2	NS (0.301) ^a^
BMI (kg/m^2^) (mean ± SD)	25.4 ± 4.5	26.3 ± 3.5	NS (0.435) ^a^
FIGO stage (N (%))			NS (0.267) ^b^
I–II	24 (80.0)	16 (66.7)	
III–IV	6 (20.0)	8 (33.3)	
Grade (N (%))			NS (0.588) ^b^
G1	8 (26.7)	4 (16.6)	
G2	18 (60.0)	15 (62.5)	
G3	4 (13.3)	5 (20.8)	
Surgical performance (N (%))			NS (0.695) ^b^
Complete	25 (83.3)	19 (79.2)	
Not complete (optimal + suboptimal)	5 (16.7)	5 (20.8)	
Neoadjuvant treatment (N (%))			NS (0.816) ^b^
Yes	2 (6.7)	2 (8.3)	
No	28 (93.3)	22 (91.7)	
Adjuvant treatment (N (%))			NS (0.462) ^b^
Yes	28 (93.3)	21 (87.5)	
No	2 (6.7)	3 (12.5)	
Recurrence (N (%))			NS (0.793) ^b^
Yes	9 (30.0)	8 (33.3)	
No	21 (70.0)	16 (66.7)	
Exitus (N (%))			NS (0.081) ^b^
Yes	7 (23.3)	11 (45.8)	
No	23 (76.7)	13 (54.2)	
OS (months) (median; Q1–Q3)	123.0; 59.5–159.0	57.5; 12.0–163.0	0.030 ^c^
DFS (months) (median; Q1–Q3)	106.0; 28.0–158.0	38.5; 10.5–163.0	NS (0.269) ^c^

BMI: body mass index; EAOC: endometriosis-associated ovarian cancer; EOC: endometrioid ovarian cancer; CCOC: clear cell ovarian cancer; FIGO: International Federation of Gynaecology and Obstetrics; SD: standard deviation; DFS: disease-free survival; OS: overall survival; NS: not significant; ^a^ Student’s *t*-test; ^b^ Pearson chi-square test; ^c^ log-rank test.

**Table 2 ijms-24-12083-t002:** The distribution of TILs in OE (*n* = 43) versus EAOC (*n* = 54) (*).

	OE (*n* = 43)	EAOC (*n* = 54)	EOC (*n* = 30)	CCOC (*n* = 24)	*p*-Value (EAOC vs. OE)	*p*-Value (EOC vs. COCC)
CD3 signal (N (%))						
Yes No	40 (100.0) 0 (0.0)	48 (98.0) 1 (2.0)	26 (96.3) 1 (3.7)	22(100.0) 0 (0.0)	NS (0.551) ^a^	NS (0.551) ^a^
Low/mild Moderate/intense	13 (32.5) 27 (67.5)	36 (78.3) 10 (21.7)	19 (79.2) 5 (20.8)	17 (77.3) 5 (22.7)	<0.001 ^a^	NS (0.876) ^b^
Intraepithelial Stromal Intraepithelial and stromal	32 (80.0) 0 (0.0) 8 (2.0)	16 (33.3) 2 (4.2) 30 (62.5)	7 (26.9) 2 (7.7) 17 (65.4)	9 (40.9) 0 (0.0) 13 (59.1)	<0.001 ^a^	NS (0.291) ^a^
CD4 signal (N (%))						
Yes No	36 (94.7) 2 (5.3)	39 (86.7) 6 (13.3)	21 (87.5) 3 (12.5)	18 (85.7) 3 (14.3)	NS (0.279) ^a^	NS (0.600) ^a^
Low/mild Moderate/intense	17 (51.5) 16 (48.5)	24 (77.4) 7 (22.6)	14 77.8) 4 (22.2)	10 (77.0) 3 (23.0)	0.031 ^a^	NS (0.955) ^b^
Intraepithelial Stromal Intraepithelial and stromal	23 (63.9) 4 (11.1) 9 (20.9)	14 (35.9) 8 (20.5) 17 (43.6)	7 (33.3) 3 (14.3) 11 (52.4)	7 (38.9) 5 (27.8) 6 (33.3)	NS (0.053) ^a^	NS (0.417) ^a^
CD8 signal (N (%))						
Yes No	36 (90.0) 4 (10.0)	34 (69.4) 15 (30.6)	20 (76.9) 6 (23.1)	14 (60.8) 9 (39.2)	0.018 ^b^	NS (0.224) ^b^
Low/mild Moderate/intense	16 (47.1) 18 (52.9)	26 (86.7) 4 (13.3)	14 (87.5) 2 (12.5)	12 (85.7) 2 (14.3)	0.001 ^b^	NS (0.886) ^b^
Intraepithelial Stromal Intraepithelial and stromal	25 (69.4) 2 (5.6) 9 (20.9)	18 (52.9) 4 (11.8) 12 (35.3)	8 (40.0) 4 (20.0) 8 (40.0)	10 (71.4) 0 (0.0) 4 (28.6)	NS (0.336) ^a^	NS (0.098) ^a^
TILs (N (%))						
Low/mild Moderate/intense	4 (11.1) 32 (88.9)	30 (73.2) 11 (26.8)			0.001 ^a^	

OE: ovarian endometriosis; EAOC: endometriosis-associated ovarian cancer; EOC: endometrioid ovarian cancer; CCOC: clear cell ovarian cancer; TILs: tumour infiltrating lymphocytes; NS: not significant. (*) Immunohistochemistry has been performed in the specimens from the whole cohort. Missing data due to technical difficulties have been excluded from the analyses. ^a^ Chi-square test. ^b^ Fisher exact test.

**Table 3 ijms-24-12083-t003:** Oncological outcome results depending on TILs signal in patients with EAOC.

	TILs Signal		
	Low/Mild	Moderate/Intense	*p*-Value
**Exitus**			
Yes	11 (33.7)	1 (9.1)	NS (0.128) ^a^
No	19 (63.3)	10 (90.9)	
**Recurrence**			
Yes	11 (63.7)	1 (9.1)	NS (0.128) ^a^
No	19 (63.3)	8 (90.9)	
**OS ≥ 68 months**			
Yes	12 (54.5)	9 (100.0)	0.030 ^a^
No	10 (45.5)	0 (0.0)	
**OS ≥ 68 months in patients with adjuvant therapy**			
Yes	10 (52.6)	9 (100.0)	0.026 ^a^
No	9 (47.4)	0 (0.0)	
**DFS ≥ 45 months**			
Yes	10 (41.7)	10 (90.9)	0.010 ^a^
No	14 (58.3)	1 (9.1)	
**DFS ≥ 45 months in patients with adjuvant therapy**			
Yes	8 (38.1)	10 (90.9)	0.008 ^a^
No	13 (61.9)	1 (9.1)	

Calculations were performed considering median overall survival and disease-free survival. DFS: disease-free survival; OS: overall survival; NS: not significant; TILs: tumour-infiltrating lymphocytes. ^a^ Fisher’s exact test.

## Data Availability

Data are available from the authors upon request.

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
