# Peer review of "Evaluation of Immune Infiltrates in Ovarian Endometriosis and Endometriosis-Associated Ovarian Cancer: Relationship with Histological and Clinical Features"

_ijms, 2023, doi:10.3390/ijms241512083_

Round 1

Reviewer 1 Report

This paper describes an statistical analysis, based  on the enumeration of several types of immune cells, identified by immunohistochemistry and/or immunofluorescence in tissue sections of Ovarian Endometriosis (OE) and  Endometriosis-associated Ovarian Cancer (EAOC).

         The described observations are of importance from the viewpoint of both basic research and clinical significance.

         The authors make statements on  the main limitations and  strengths of their paper.

Specific comment

         While the meaning of the examined cell subsets is understandable, throughout the whole paper, some terms and descriptions are rather confusing.

         For  instance, in line 39,  as said “exhaustion markers (PD-1, TIM3, CD39, FOXP3 and CD163)”  seem to include CD163 as an exhaustion marker…

         In line 275  Regarding immune checkpoint expression, FOXP3+/CD39+/CD163+…”  seem to be considered as immune checkpoints while somewhere else they are described as T-cell exhaustion markers…

         The authors should make an effort, in describing and explaining in a better way, the meaning and cell distribution of those immune terms and markers.

Author Response

Response: We thank the reviewer for the opportunity of clarifying some aspects of our manuscript, which allowed us to increase its quality. As reported in the literature, some immune checkpoints (i.e. PD-1 and TIM3) could be considered, together with other markers (such as CD39) as markers of T-cell exhaustion (Balança et al, 2020; PMID: 33332284, reference [29] of the manuscript). Following the reviewer's thoughtful suggestion, we have incorporated the following changes to increase the comprehension of our work: Lines 38 to 40: the sentence “TILs (CD3+,CD4+,CD8+) and macrophages (CD163) were assessed by immunochemistry and exhaustion markers (PD-1, TIM3, CD39, FOXP3 and CD163) by immunofluorescence on paraffin-embedded samples from n=43 OE and n=54 EAOC patients” has been modified by the more accurate expression “Exhaustion markers (PD-1, TIM3, CD39, FOXP3) and their relationship with macrophages (CD163) were assessed by immunofluorescence on paraffin-embedded samples from n=43 OE and n=54 EAOC patients.” Lines 91 to 93: the expression “the later being in turn characterized by the expression of multiple immune checkpoints as CD39, PD-1, and TIM3” has been replaced by the more precise expression “the latter being in turn characterized by the expression of markers as CD39 and immune checkpoints as PD-1 and TIM3). Line 96 to 97: the sentence “The primary objective of this study is to identify and compare the spatial distribution of CD3+/CD4+ and CD3+/CD8+ TILs and macrophages in OE and EAOC” has been improved with the following sentence “The primary objective of this study is to identify and compare the spatial distribution of CD3+/CD4+ and CD3+/CD8+ TILs and tumor-associated macrophages (TAMs) in OE and EAOC.” Line 171: the expression “distribution of CD163+ macrophages” has been replaced by “distribution of CD163+ TAMs”. Line 264: the sentence “Immunofluorescence staining for T-cell exhaustion markers was performed in those EAOC specimens(...)” has been improved as “Immunofluorescence staining for T-cell exhaustion and TAMs markers were performed in those EAOC specimens(...) Line 275: the expression “Regarding immune checkpoint expression, FOXP3+/CD39+/CD163+...” has been replaced by the expression “Regarding the expression of T-cell exhaustion and TAMs markers,...”.

Reviewer 2 Report

Esteemed authors and editorial team,

This is a very interesting research targeting the potential link between endometriosis and ovarian cancer and moreover, offering a possible insight to an immunologic stamp of cases with potential malignant degenerescence.

I consider the manuscript amenable for publications following minor changes.

Please check that all abbreviations are explained at the first mention in the text, for example OS and DFS are not explained in the abstract.

Another suggestion is to take into discussion the potential clinical applicability of findings.

Author Response

Response: We thank the reviewer for pointing out this inaccuracies of our manuscript. Accordingly, we have modified the following sentences:

Line 43: the abbreviations “OS” and “DFS” have been modified by “overall survival” and “disease-free survival”, respectively.

Line 211: we have incorporated “body mass index” to explain the abbreviation BMI.

Page 6, table 2: “IHC” has been replaced by “immunohistochemistry” for clarification purposes.

Regarding the clinical applicability of the findings, the following paragraph has been added in line 374:

The results of the current study confirmed the presence of a subset of exhausted T cells in EAOC, which regarding OC, had only been described in the high-grade serous subtype, to the best of our knowledge. This might represent an opportunity to select those patients which could benefit the most from immunotherapy. Regarding EAOC, this could become a specially attractive therapeutic opportunity CCOC, the subtype of OC more resistant to conventional chemotherapy.
